



# Untangling causality in midlatitude aerosol-cloud adjustments

Daniel T. McCoy[1], Paul Field[1,2], Hamish Gordon[1], Gregory S. Elsaesser[3], Daniel P. Grosvenor[1,4]

[1]Institute of Climate and Atmospheric Sciences, University of Leeds, UK
[2]Met Office, UK
[3]Department of Applied Physics and Applied Mathematics, Columbia University and NASA Goddard Institute for Space Studies, New York, NY, USA
[4]National Centre for Atmospheric Science, Leeds, UK

*Correspondence to*: Daniel T. McCoy (D.T.McCoy@leeds.ac.uk)

**Abstract.** Aerosol-cloud interactions represent the leading uncertainty in our ability to infer climate sensitivity from the
observational record. The forcing from changes in cloud albedo driven by increases in cloud droplet number ($N_d$) (the first indirect effect) is confidently negative and has narrowed its probable range in the last decade, but the sign and strength of forcing associated with changes in cloud macrophysics in response to aerosol (aerosol-cloud adjustments) remain uncertain. This uncertainty reflects our inability to accurately quantify variability not associated with a causal link flowing from the cloud microphysical state to cloud macrophysical state. Once variability associated with meteorology has been removed,
covariance between the liquid water path averaged across cloudy and clear regions (LWP, here, characterizing the macrophysical state) and $N_d$ (characterizing the microphysical) is the sum of two causal pathways linking $N_d$ to LWP: $N_d$ altering LWP (adjustments) and precipitation scavenging aerosol and thus depleting $N_d$. Only the former term is relevant to constraining adjustments, but disentangling these terms in observations is challenging. We hypothesize that the diversity of constraints on aerosol-cloud adjustments in the literature may be partly due to not explicitly characterizing covariance
flowing from cloud to aerosol, and aerosol to cloud. Here, we restrict our analysis to the regime of extratropical clouds outside of low-pressure centers associated with cyclonic activity. Observations from MAC-LWP, and MODIS are compared to simulations in the MetOffice Unified Model (UM) GA7.1 (the atmosphere model of HadGEM3-GC3.1 and UKESM1). The meteorological predictors of LWP are found to be similar between the model and observations. There is also agreement with previous literature on cloud-controlling factors finding that increasing stability, moisture, and sensible heat flux
enhance LWP, while increasing subsidence, and sea surface temperature decrease it. A simulation where cloud microphysics are insensitive to changes in $N_d$ is used to characterize covariance between $N_d$ and LWP that is induced by factors other than aerosol-cloud adjustments. By removing variability associated with meteorology and scavenging we infer the sensitivity of LWP to changes in $N_d$. Application of this technique to UM GA7.1 simulations reproduces the true model adjustment strength. Observational constraints developed using simulated covariability not induced by adjustments and observed
covariability between $N_d$ and LWP predict a 25-30% overestimate by the UM GA7.1 in LWP change and a 30-35% overestimate in associated radiative forcing.





## 1 Introduction

Uncertainty in the radiative forcing due to aerosol-cloud interactions is the leading uncertainty limiting our ability to accurately diagnose the Earth's climate sensitivity from the observational record (Forster, 2016). The best estimate of the radiative forcing due to aerosol-cloud interactions (also called the first indirect effect (Twomey, 1977)) has narrowed to -1.2 to -0.34 Wm$^{-2}$ in a recent survey of forcing from aerosol-cloud interactions (Storelvmo, 2017; Bellouin et al., 2019), but the sign and strength of the forcing due to changes in cloud macrophysical properties in response to aerosol (aerosol-cloud adjustments) remain uncertain (Bellouin et al., 2019). This uncertainty reflects the difficulty in disentangling the many factors that determine cloud macrophysical properties. Unlike cloud droplet number concentration ($N_d$), which is primarily driven by the availability of suitable aerosol, cloud macrophysical properties are primarily determined by the state of the atmosphere, but may be modulated by $N_d$ (Stevens and Feingold, 2009).

Here, we focus on liquid clouds. Within liquid clouds two main processes are hypothesized to alter cloud liquid content in response to changes in $N_d$. As $N_d$ is increased there may occur (1) a suppression of precipitation by enhanced $N_d$ (Albrecht, 1989; Pincus and Baker, 1994), and (2) strengthened entrainment of dry air (Ackerman et al., 2004; Bretherton et al., 2007; Wang et al., 2003; Xue and Feingold, 2006). These processes drive the liquid content of cloud in opposite directions following a perturbation in microphysics, further complicating the interpretation of covariability between cloud macrophysical and microphysical properties. Because these processes exist at a time and length scale far below those resolved in climate models they must be parameterized, resulting in substantial uncertainty. Because of this constraining the cloud macrophysical response to changes in cloud microphysics using observations is an essential step towards constraining aerosol forcing. Aerosol-cloud interactions are likely to contribute a relatively small fraction of overall variability in cloud properties- making the data volume available from remote-sensing observations particularly relevant to providing an observational constraint. The literature has produced numerous careful analyses of the observational record, but these analyses have produced divergent estimates of the aggregate effect of increased $N_d$ on liquid clouds. These impacts range from increased $N_d$ increasing liquid content in clouds (Chen et al., 2014; Rosenfeld et al., 2019), to almost no response in in-cloud and area-averaged LWP (Toll et al., 2017; Malavelle et al., 2017), to decreasing liquid content (Gryspeerdt et al., 2019; Sato et al., 2018). In contrast, the response in cloud cover to $N_d$ has tended to be consistently inferred as producing a forcing equal or larger than the first indirect effect (Gryspeerdt et al., 2016; Christensen et al., 2017; Andersen et al., 2017). Simulation of deep open cellular boundary layers has shown an increase in cloud fraction, and a decrease in in-cloud LWP, ultimately resulting in an increase in reflected shortwave in the simulation (Possner et al., 2018). Given the wide range of potential observational constraints it remains difficult to offer advice on what cloud microphysical parameterizations are the most realistic.

However, the causal link flowing from $N_d$ to clouds is not the only mechanism relating cloud macrophysics to cloud microphysics. As shown in Wood et al. (2012), the spatial pattern in $N_d$ observed in nature is primarily determined by precipitation scavenging. That is to say, $N_d$ may both alter cloud macrophysical properties, and be altered by them and




interpreting observations as a constraint on only microphysical to macrophysical causality is erroneous, as discussed in Gryspeerdt et al. (2019). To be able to constrain cloud responses to aerosol-driven changes in $N_d$ we must be able to characterize the effects of cloud macrophysical properties on $N_d$ (by precipitation scavenging) and covariability between $N_d$ and LWP induced by confounding factors such as the relative location of aerosol sources and climatological cloud cover, or

meteorological modulation of aerosol and clouds.

       Bender et al. (2019) proposed utilizing the observed covariance between $N_d$ and LWP as an aggregate measure of aerosol-cloud-radiation behavior across climate models and reanalysis. That is to say, for a model to be able to realistically reproduce aerosol-cloud behavior, one precondition is that covariance flowing from clouds to aerosol, and from aerosol to clouds as well as the non-causal covariance induced by airmass history must result in a total covariance consistent with

observations. Similarly, Gryspeerdt et al. (2019) examined covariance between $N_d$ and in-cloud LWP, but proposed to separate variability induced by adjustments from other sources of covariability by using clustering on global data. This study follows on from these earlier studies and attempts to partition covariance between LWP and $N_d$ related to causality flowing from $N_d$ to LWP from other covariance. We utilize empirical analysis of observations and model output to try and disentangle variability driven by meteorological 'cloud-controlling factors' (Stevens and Brenguier, 2009) and variability

related to changes in cloud microphysics. Within this framework we focus on the midlatitudes. Previous work has performed similar analysis on cyclonic midlatitude systems, finding a distinct increase in LWP at a fixed precipitation rate (McCoy et al., 2018c). These large synoptic systems account for roughly half the midlatitudes (Bodas-Salcedo et al., 2014). Here, we turn our attention to the remaining interstitial regions between cyclones. This regime tends to be less cloudy, have an overall lower albedo, and lower cloud optical depth (an example is shown in Fig. 1a). Thus, while these regions do not host dramatic

fronts and cloud shields, changes in their liquid cloud due to anthropogenic aerosol may contribute strongly to the overall aerosol forcing because cloud areal coverage and cloud optical depth are farther below saturation.

## 2 Methods

### 2.1 Region and variable selection

25       The empirical analysis on both observational and simulated data in this work follows pioneering studies investigating meteorological controls on subtropical (Myers and Norris, 2013, 2015; Qu et al., 2015; Klein et al., 2017; Seethala et al., 2015) and midlatitude (Miyamoto et al., 2018) marine boundary-layer cloudiness. The selection of variables used in this study is based on the findings of these earlier studies. The predictors examined here are sea surface temperature (SST), large scale subsidence ($\omega$) at 550 hPa, the strength of the inversion at the top of the boundary layer (estimated

inversion strength, EIS, as defined in Wood and Bretherton (2006)), the total water vapor path (WVP), and sensible heat flux (SHF). A summary is given in Table 1. A priori we expect that increasing SST should lead to decreasing cloudiness (Myers and Norris, 2015; Qu et al., 2015; McCoy et al., 2017; Bretherton and Blossey, 2014). Increasing subsidence should decrease



liquid water path (Myers and Norris, 2013). Increasing inversion strength should lead to increasing cloud fraction (Wood and Bretherton, 2006; Klein and Hartmann, 1993; Qu et al., 2015; Myers and Norris, 2015; Seethala et al., 2015). As shown in Miyamoto et al. (2018) from examination of midlatitude ocean fronts, increased SHF should increase cloudiness. Finally, while less extensively studied, an overall increase in WVP seems likely to lead to increased cloudiness as the atmospheric

column moisture increases, holding all else equal (McCoy et al., 2018b).

        The central goal of this study is to develop an empirical constraint on aerosol-cloud adjustments. However, direct measurements of the aerosol state to constrain aerosol-cloud adjustments have been found to be sensitive to errors in pristine conditions (Ma et al., 2018). To develop a constraint on aerosol-cloud adjustments we examine the cloud microphysical state in addition to the meteorological parameters listed above. Cloud droplet number concentration is the state variable relating

cloud macrophysics to aerosol concentration (Wood, 2012; Grosvenor et al., 2018) and is the predictor used in this study to characterize aerosol-cloud adjustments.

        We hasten to note that the list of predictors described above is not intended to be complete (that is to say, we do not anticipate that we will be able to explain 100% of the variability in clouds with these predictors). Our goal is explain enough variability with these predictors to be able to infer statistically-robust relationships between cloud macrophysical and

microphysical properties in the variance unexplained by meteorological variability. The predictive abilitiy of these inferred relationships will be tested within the context of a global climate model (GCM).

        Having discussed predictors, we will now discuss predictands. Out of cyclone cloud cover is primarily liquid, boundary layer cloud, which substantially affects shortwave radiation, but has little effect on longwave radiation (Hartmann and Short, 1980). Thus, we focus on liquid water path (LWP) averaged over cloudy and clear regions. This variable allows

observations and model output to be compared without needing to simulate output from a passive spectoradiometer (Bodas-Salcedo et al., 2011) or considering cloud overlap, as would be required for in-cloud LWP .

        Analysis is carried out in the northern hemisphere (here 30°-60°N) over oceans. This choice has been made for several reasons: First, there is a large variability in $N_d$ across the northern midlatitudes that is not strongly driven by the seasonal cycle. In the Southern Ocean cloud condensation nuclei (CCN) variability is primarily due to biogenic sources and

their variability is strongly driven by the seasonal cycle, making analysis in the context of meteorological controls difficult to interpret (Ayers and Gras, 1991; McCoy et al., 2015a; Charlson et al., 1987). Second, Southern Ocean $N_d$ is still poorly represented in the MetOffice Unified Model GA7.1 (discussed in section 2.4) with both too low a mean value and generally too small a seasonal cycle – in contrast to the Northern Hemisphere, which compares well with observations (Mulcahy et al., 2018).


## 2.2 Outside of cyclone-compositing

The passage of synoptic systems is the central mode of variability in midlatitude cloudiness. McCoy et al. (2018c) focused on cyclone systems and showed that once meteorological variability was accounted for the effects of changes in cloud





microphysics driven by changes in $N_d$ on liquid content are revealed. This work utilized the Field and Wood (2007) cyclone compositing algorithm, which uses sea level pressure (SLP) to identify cyclone centers. Here we utilize the same compositing approach, but focus on all data that is 2000 km away from a cyclone center, following Bodas-Salcedo et al. (2014). This separates the midlatitudes into times when a cyclone is nearby and when there is no cyclone nearby. Once

cyclone centers are identified all data within 2000km of an identified cyclone center is masked out of the data set. All data over land is also masked because microwave observations are unavailable (see next section).

A schematic representation of the frequency of occurrence of out-of-cyclone states is shown in Fig. 1b and is 42% in the 30°N-60°N region focused on in this study. All data is regridded to a common 1°x1° spatial resolution before analysis. All data is daily-mean, except for observed $N_d$, which is only available for the MODIS Aqua overpass; however the diurnal

cycle of $N_d$ is thought to be relatively slight (Dong et al., 2014) and it is unlikely that this retrieval limitation significantly impacts our results.

## 2.3 Observations

Cloud liquid water path (LWP) is calculated based on the aggregated observations from multiple satellite microwave sensors following the methodology of the multisensory advanced climatology of LWP (MAC-LWP; Elsaesser et

al. (2017)) at daily-mean resolution. Microwave radiometers are sensitive to total liquid. This means that the retrieval must calculate a partitioning between precipitating and non-precipitating liquid. Here, the liquid water path in clouds is calculated following Elsaesser et al. (2017). Previous analysis has shown that the midlatitude partitioning or rain and cloud in MAC-LWP compares favorably with convection-permitting simulations (McCoy et al., 2018c). The MAC-LWP data set estimates that 7% of total liquid path is rain in the NH midlatitudes outside of cyclones. Because of the low fraction of rain water in

the midlatitudes outside of cyclones it is unlikely that the need to partition the observations into rain and cloud water substantially affects observational constraints as calculated in section 3.3. Liquid water path is defined as the average of cloudy and clear regions and is insensitive to overlying ice cloud (unlike passive optical retrievals). This definition of LWP is consistent with the definition from GCMs. Microwave retrievals are only available over ocean.

Observations of $N_d$ are calculated based on MODIS observations of cloud optical depth and droplet effective radius.

Level-2 swath data (joint product) from MODIS collection 5.1 (King et al., 2003) is filtered by removing pixels with solar zenith angles greater than 65° to eliminate  problematic retrievals at a pixel-level (Grosvenor and Wood, 2014). The daily-mean $N_d$ at 1°x1° resolution is calculated from the filtered level 2 swath data and only low (cloud tops below 3.2 km), liquid clouds were used to calculate $N_d$. Only 1°x1° regions where the liquid cloud fraction exceeds 80% are considered valid (Bennartz et al., 2011) and the $N_d$ is calculated using effective radius from the  3.7μm MODIS channel. This data set is

evaluated in Grosvenor et al. (2018) and in McCoy et al. (2018a), where it shows consistency with measurements from aircraft.

We feel that this combination of radiometers in constraining adjustments is particularly advantageous because there is no shared information between the microwave retrieval of LWP, and the shortwave and near-infrared retrieval of $N_d$. This





is in contrast to the common practice of utilizing MODIS retrievals of cloud droplet effective radius and optical depth to calculate both the in-cloud LWP and $N_d$, making it conceptually difficult to cleanly infer changes in LWP due to changes in $N_d$.

The meteorological state of the atmosphere is characterized using reanalysis from MERRA2 (Molod et al., 2015)
and microwave observations of water vapor path (WVP) from MAC-LWP. The period during which all of these observations are available stretches from 2003 to 2015. A list of variables and data sources is given in Table 1.

**2.4 Simulations**

Simulations are carried out in atmosphere-only simulations in the MetOffice Unified (UM) model version 10.8 at N96 horizontal resolution. The version of the atmosphere model is GA7.1 coupled to the UKCA aerosol model as described
in (Mulcahy et al., 2018). The GA7.1 atmosphere model is the physical basis for the atmosphere model included in the HadGEM3-GC3.1 and UKESM1 climate model submission to CMIP6 (Walters et al., 2019). All simulations are two and a half years starting in September of 2013. Simulations are nudged to European Centre for Medium-Range Weather Forecasts (ECMWF) reanalysis winds above 1.7 km. Paired simulations are conducted with aerosols set to pre-industrial (henceforth labeled PI) and to present day levels (labeled PD). These simulations are conducted to calculate the response of cloud to
anthropogenic aerosol and are nudged to temperature and wind above 1.7 km.

In GA7.1 aerosol number concentration has the potential to affect cloud microphysics in three places: First, the conversion of cloud water to rain, where enhanced $N_d$ suppresses the conversion of cloud water to rain water. This is parameterized using the scheme in Khairoutdinov and Kogan (2000), bias corrected as described in (Boutle et al., 2014). The conversion rate of cloud to rain is parameterized as

$$P_{Cloud \rightarrow Rain} \propto q_{Cloud}^{2.47} \cdot N_d^{-1.79} \qquad [1]$$

where $q_{Cloud}$ is the cloud liquid water mixing ratio. Second, the gravitational settling of cloud droplets, where enhanced $N_d$ suppresses the settling of cloud droplets out of the cloud. The flux of water settling out of a cloud layer is given as

$$P_{Settle} \propto q_{Cloud}^{5/3} N_d^{-2/3} \qquad [2]$$

Third, enhanced cloud albedo from changes in $N_d$ in the radiative transfer code (Twomey, 1977).

However, clouds may affect aerosol concentrations via wet scavenging. UKCA allows scavenging to be configured in two ways: the default configuration, or as in the ECHAM5-HAM model (Stier et al., 2005). To evaluate some of the uncertainty related to scavenging we perform simulations with both sets of coefficients.  A list of model simulations conducted is given in Table 2.

**2.5 Analyzing covariance in the system**

Our goal in this paper is to characterize how $N_d$ drives LWP. However, to reveal robust correlations between $N_d$ and LWP, meteorological variability must be accounted for. To achieve this we follow previous empirical analysis of cloud controlling



factors and utilize multiple linear regression to characterize the dependence of various cloud properties as predictands on the predictors described above (Myers and Norris, 2015). As with all empirical analyses of observations, correlation between predictors and the predictand does not necessarily mean that they are causally linked. The meteorological predictors selected here have had mechanisms hypothesized to link them to cloud variability. Another issue with this, and similar analyses of the

response of cloud to cloud-controlling factors is that thermodynamic and dynamical predictors within the Earth's atmosphere are correlated (McCoy et al., 2017; Myers and Norris, 2015, 2013). As the variance shared by predictors grows, the uncertainty in the coefficients calculated by multiple linear regression increases. While this is an issue, it is inescapable in trying to disentangle the effects of different cloud controlling factors on cloud properties. Examples of meteorological variables that are correlated, but have opposing effects on cloud cover include subsidence and stability; and stability and SST

(Myers and Norris, 2013; Qu et al., 2015).

To reduce predictor covariance we bin our 1°x1° daily-mean data from model output and observations into the space of SST and WVP. All output over oceans is considered in the analysis. We choose these predictors to bin our data by before performing regression analysis because they represent two of the most basic meteorological state variables, they covary strongly, and because LWP varies significantly as a function of WVP and SST in both observations and simulations

(Fig. 2).   Nine bins are used to span WVP (0-40 kg/m2) and ten for SST (275-305 K). We propose that a multiple linear regression model of the form

$$LWP = a_{ccn}CCN + a_2\omega_{550} + a_3EIS + a_4WVP + a_5SST + a_6SHF + a_7 \qquad [3]$$

be trained in each bin of SST and WVP. All the predictors within the regression model are environmental drivers that are external to the clouds being influenced. However, robust remote-sensing observations of CCN are lacking and $\ln(N_d)$ is used

in its place. As discussed above, $N_d$ is predominantly a function of CCN availability. Thus, the multiple linear regression model

$$LWP = a_1\ln(N_d) + a_2\omega_{550} + a_3EIS + a_4WVP + a_5SST + a_6SHF + a_7 \qquad [4]$$

is trained in each bin of WVP and SST. Bins that contain fewer than 3000 data points are excluded. Units of LWP are in g m$^{-2}$, $N_d$ is in cm$^{-3}$, subsidence is in Pa s$^{-1}$, EIS is in K, WVP is in kg m$^{-2}$, SST is in K, and SHF is in Wm$^{-2}$. A 99% confidence

interval is used to determine if a coefficient is significant. The initiation of precipitation in boundary layer cloud can substantially impact LWP within a given cloud with a timescale longer than the daily sampling time scale utilized here (Berner et al., 2013). However, the spatial scale of our analysis (1°x1°) samples clouds at many stages of their lifecycle. This means we do not need to consider rain as a predictor of LWP, and may think of it as scaling with cloud LWP.

Training the regression model in Eq. 4 produces a measure of the covariance between predictors and predictand. In

particular, it produces a measure of the covariability between $\ln(N_d)$ and LWP ($a_1$). However, this covariance does not show causality. $N_d$ may enhance LWP by, for instance, suppressing rain (Albrecht, 1989), and it is this relationship where causality runs from $N_d$ to LWP that is of interest in understanding adjustments. However, the observed covariability between $N_d$ and LWP is also the product of causality running from LWP to $N_d$ due to precipitation removing aerosol and reducing $N_d$ (Wood et al., 2012). The overall covariance between $N_d$ and LWP is the product of these two causal pathways as well as





correlation induced by environmental factors driving both clouds and aerosol, and by geographic and seasonal distribution of sources of aerosol relative to cloud.

How do we constrain adjustments based on examination of modeled and observed $N_d$ and LWP if we can't disentangle causality? The solution we propose is to use a simplified version of the model to characterize the sensitivity of
$N_d$ to precipitation and other factors and use this sensitivity to interpret the full model. To conceptualize why this process can estimate the sensitivity of LWP to $N_d$ we can write an equation describing small changes in LWP in the space of small changes in $N_d$. First, we consider $N_d$ acting on cloud microphysics ($N_{d_{Cloud}}$) and $N_d$ when precipitation scavenging is the only causal factor linking cloudiness and $N_d$ ($N_{d_{Scav}}$) as separate entities and consider changes in LWP relative to these two quantities. That is to say we consider $LWP = fn(\ln(N_{d_{Cloud}}), \ln(N_{d_{Scav}}))$, where the logs are added for consistency with
Eq. 4 and all other terms in equation 1 are held fixed and the first term is expanded.

Conceptually, LWP is increased with increasing $N_{d_{Cloud}}$ (Fig. 3b) and $N_{d_{Scav}}$ is reduced by increasing rain-rates, which increase with LWP (Fig. 3c). In the topographic space describing LWP in terms of $N_{d_{Cloud}}$ and $N_{d_{Scav}}$, reduced $N_{d_{Scav}}$ corresponds to increased precipitation rates, and in turn, LWP. That is to say

$$\frac{dN_{dScav}}{dR} < 0, \quad \frac{dR}{dLWP} > 0 \tag{5}$$

where R is rain rate, so the reciprocal of the product of these terms gives

$$\frac{dLWP}{dN_{dScav}} < 0 \tag{6}$$

Thus, a small change in LWP in terms of changes in these two variables alone is

$$dLWP = \frac{\partial LWP}{\partial \ln(N_{d_{Scav}})} d\ln(N_{d_{Scav}}) + \frac{\partial LWP}{\partial \ln(N_{d_{Cloud}})} d\ln(N_{d_{Cloud}}) \tag{7}$$

We assume that scavenging is unaffected by changes in $N_{d_{Cloud}}$ affecting rain rates, so there are no higher-order
terms, although this may not be the case in reality (Wood et al., 2012), and this feedback is found to substantially enhance the strength of adjustments in some models (Jing and Suzuki, 2018). A visualization of Eq. 7 is shown in Fig. 3a. In the real world $N_d = N_{d_{Cloud}} = N_{d_{Scav}}$ and the best fit of LWP to $N_d$ gives the sum of the partial derivatives of the two terms following

$$\frac{dLWP}{d\ln(N_d)} = \left(\frac{\partial LWP}{\partial \ln(N_{d_{Scav}})}\right)_{N_{dCloud}} \frac{d\ln(N_{d_{Scav}})}{d\ln(N_d)} + \left(\frac{\partial LWP}{\partial \ln(N_{d_{Cloud}})}\right)_{N_{dScav}} \frac{d\ln(N_{d_{Cloud}})}{d\ln(N_d)} \tag{8}$$

where subscripts on the partial derivatives indicate that they are evaluated holding that variable constant.

$$\frac{dLWP}{d\ln(N_d)} = \left(\frac{\partial LWP}{\partial \ln(N_{d_{Scav}})}\right)_{N_{dCloud}} + \left(\frac{\partial LWP}{\partial \ln(N_{d_{Cloud}})}\right)_{N_{dScav}} \tag{9}$$

We always sample along the dashed line in Fig. 3a in the observational record or in a model where clouds are sensitive to $N_d$ and aerosol is scavenged, giving the curve in Fig. 3d. This curve combines the effects of causality flowing from $N_d$ to LWP, and from LWP to precipitation to aerosol and onto $N_d$. The term $\frac{\partial LWP}{\partial \ln(N_{d_{Cloud}})}$ is the key to constraining





adjustments because it describes the sensitivity of LWP to changes in $N_d$ (in this analogy the slope of LWP along the red line in Fig. 3a). While we cannot disentangle causality in observations or in a model where $N_d$ affects cloud microphysics, we propose that in a model configuration where cloud microphysics (e.g. radiation, settling, and autoconversion in UM GA7.1) are unaffected by $N_d$ we may estimate the term $\frac{\partial LWP}{\partial \ln (N_{d_{Scav}})}$. That is to say, if $N_{d_{Cloud}} = constant$ and $N_d = N_{d_{Scav}}$ then

$$\frac{dLWP}{d\ln(N_d)} = \left(\frac{\partial LWP}{\partial \ln (N_{d_{Scav}})}\right)_{N_{dCloud}} \qquad [10]$$

and the best fit of LWP to $N_d$ in the scavenging-only simulation is just a measure of the scavenging of aerosol by precipitation and the non-causal covariance between $N_d$ and LWP induced by other confounding factors.

Thus, the sensitivity $\frac{\partial LWP}{\partial \ln (N_{d_{Cloud}})}$ can be estimated as the difference between the regression of LWP on $N_d$ trained in a simulation where clouds are insensitive to $N_d$ ($N_{d_{Cloud}} = constant$, scavenging-only) and the regression in a control

simulation ($N_d = N_{d_{Cloud}} = N_{d\,Scav}$) because

$$\left(\frac{\partial LWP}{\partial \ln (N_{d_{Cloud}})}\right)_{N_{dScav}} = \frac{dLWP}{d\ln(N_d)} - \left(\frac{\partial LWP}{\partial \ln (N_{d_{Scav}})}\right)_{N_{dCloud}} \qquad [11]$$

We will use this correction for variability induced by factors besides adjustments throughout the paper to infer the effects of $N_d$ on cloud properties in model configurations and the observations.

To reiterate, non-causal factors also lead to correlation between $N_d$ and LWP because air mass history is important

for both clouds and aerosol (Mauger and Norris, 2007). In the discussion of the conceptual model presented above we focused on the impact of changes in $N_d$ on LWP, and vice-versa. Spurious correlation due to air mass history will affect both terms on the right-hand side of Eq. 11 when they are calculated using regression. Thus, the correction for scavenging-induced variability in Eq. 11 also corrects for variability induced by air mass history simultaneously affecting $N_d$ and LWP.

## 3 Results

### 3.1 Variance in LWP as a function of $N_d$

As described in the methods section, data from observations and the control simulation in the UM between 30°N and 60°N over oceans and 2000 km from cyclone centers at daily-mean resolution is binned by WVP and SST and the regression model shown in Eq. 4 is trained in each bin. If we plot the quantity

$$LWP - a_2\omega_{500} - a_3EIS - a_4WVP - a_5SST - a_6SHF - a_7 \qquad [12]$$

as a function of $N_d$ we can see that there is significant variability in LWP associated with $N_d$, once variability associated with meteorological predictors is removed (Fig. 4). The residual variance in LWP covaries with $N_d$ similarly between the observations and UM simulations where observations exist. However, one key difference is the span of $N_d$ values in the observations and simulations. Observed $N_d$ almost never falls below 30 cm$^{-3}$ (the dashed line in Fig. 4), while the $N_d$ calculated by the model occasionally falls below this value (~10% of modeled $\ln(N_d)$). This is not surprising because the





model suffers from no retrieval limitations and can always measure an $N_d$, even when cloud is extremely tenuous. Ultimately the goal of this study is to utilize the slope of LWP with respect to $N_d$ to infer adjustment strength. However, using the slope from the observations and the models will produce quite different values due to the strong increase in LWP with $N_d$ below 30 cm$^{-3}$. In order to compare observations and models fairly we exclude data for Nd<30 cm$^{-3}$.

For $N_d$>30 cm$^{-3}$ residual LWP decreases with increasing $N_d$ in both observations and models. This agrees with the notion that precipitation variability drives $N_d$ (Wood et al., 2012), and agrees with previous analysis (Gryspeerdt et al., 2019). Clearly, this does not mean that the model reduces cloudiness in response to aerosol. The cloud microphysics within the UM GA7.1 must increase LWP in response to a change in $N_d$, and analysis of the model response to changes from pre-industrial (PI) to present day (PD) aerosol have confirmed a negative forcing from adjustments (Mulcahy et al., 2018). As

discussed in the methods, the difference in slopes between the scavenging-only and control simulation is a proxy for the response in cloud microphysics to $N_d$. Before analyzing this difference, we will briefly discuss the covariance between LWP and meteorological predictors in the observations and the control simulations.

**3.2 Covariances between LWP and meteorology**

Before returning to discussing aerosol-cloud adjustments, we discuss the covariances between observed LWP and the other

predictors in Eq. 4. This is shown in Fig. 5. The explained variance ($R^2$) by the regression model exceeds 40% except at low WVP and SST. The variance explained across all bins of SST and WVP is 46% of daily-mean 1x1° variability. That is to say, nearly half the day-to-day variability in LWP across the midlatitudes away from cyclones can be explained as a simple linear combination of five variables.

       Several of the predictors have uniform effects on LWP: LWP decreases with subsidence (consistent with Myers and

Norris (2013)); increases with WVP; decreases with increasing SST (consistent with Qu et al. (2015), and references therein); and increases with sensible heat flux (consistent with Miyamoto et al. (2018)). We note that at very low SST it appears that LWP may increase with SST. It is possible that this feature is related to ice to liquid transitions (McCoy et al., 2015b; McCoy et al., 2016; Tsushima et al., 2006; Senior and Mitchell, 1993; Tan et al., 2016), but this region only accounts for a small fraction of the overall data volume. The only meteorological predictor that correlates positively and negatively

with LWP across the SST-WVP phase space is EIS. At higher WVP and lower SST EIS and LWP are negatively correlated, while at low WVP and high SST they are positively correlated. The latter effect is consistent with increasing EIS increasing cloud fraction (Wood and Bretherton, 2006). One possibility is that because clouds are closer to 100% in areal coverage at higher WVP and lower SST, increasing stability suppresses thickening of cloud in this regime, while the clouds cannot expand horizontally.

Examination of the control simulation shows very similar covariances between meteorological predictors and LWP (Fig. 6). Notable differences are the stronger positive covariance between SST and LWP at low SST, and uniform negative covariance between EIS and LWP across the SST-WVP phase space. However, the relationship between LWP and meteorology is strikingly similar between observations and the UM. The explained variance by predictors tends to be higher





in the UM GA7.1 (64%), but it is hard to say how much of this difference in explained variance is due to simplification of the real atmosphere and cloud physics by the model, and how much is due to observational error.

In this framework we also characterize the covariance between $\ln(N_d)$ and LWP. The correlation between $\ln(N_d)$ and LWP is primarily negative in the observations, with positive correlations only occurring at high WVP and low SST. The correlation between LWP and $\ln(N_d)$ in the model is almost uniformly negative (in the data set restricted to $N_d$>30 cm-3). Standardization of the predictor strength estimates the effect of a standard deviation change in each predictor in standard deviations of LWP. In both the observations and models the impact of a standard deviation in meteorological predictors dwarfs the effect of a standard deviation in $\ln(N_d)$, with contributions from subsidence and SHF dominating variability. That is to say, the relative contributions of variability in $N_d$ are quite small in comparison to variability that is simply due to the weather.

### 3.3 Inferring the effect of $N_d$ on LWP

As we have seen, the correlation between LWP and $N_d$ is mostly negative. The slope relating $\ln(N_d)$ to LWP in the context of Eq. 4 is reproduced in Fig. 7ab. However, if we manually set the $N_d$ seen by autoconversion, radiation, and settling to 75 cm$^{-3}$ in the UM GA7.1 (the approximate mean value in the study region) the negative covariance strengthens substantially (Fig. 7c). This model version is termed 'scavenging-only' because scavenging is the only causal link between cloud and aerosol. As discussed in the methods section, the effect of changes in $N_d$ on LWP may be approximated by the difference between Fig. 7b and Fig. 7c. This is shown in Fig. 7e. This agrees with our expectation based on our knowledge of how the UM GA7.1 treats liquid cloud processes. We know that the effect of increasing $N_d$ on autoconversion and settling is to inhibit the conversion of cloud to rain and reduce cloud droplet fall speed.

To evaluate the accuracy of this method in inferring the sensitivity of LWP to $N_d$ we conduct simulations where causality is forced to flow from the $N_d$ to LWP. This is done by setting the $N_d$ seen by the autoconversion, settling, and radiation to constant values. Two simulations are conducted for each mechanism with $N_d$ set to 100 cm$^{-3}$, and 300 cm$^{-3}$. The $N_d$ seen by the other mechanisms is held constant at 30 cm$^{-3}$. A further three simulations are conducted where $N_d$ seen by all three mechanisms is set to 30 cm$^{-3}$, 100 cm$^{-3}$, and 300 cm$^{-3}$ (Table 2). This yields a set of simulations describing the effects on LWP of an increase in $N_d$ in the autoconversion, settling, or radiation, as well as via all three mechanisms simultaneously. Analysis of the effects of increased $N_d$ seen by the radiation code showed negligible changes in LWP and are not shown here.

As in Fig. 6, the fit in Eq. 4 is trained in each bin of WVP and SST across each set of three simulations. The $N_d$ seen by the model parameterization is used in the multiple linear regression (Eq. 4). As expected, given the functional form of these parameterizations (Eq. 1 and Eq. 2) the effect of increasing $N_d$ in either autoconversion or settling results in an increase in LWP, as does the case where $N_d$ in the settling, autoconversion, and radiation are all varied together. The effect of a





perturbation in the $N_d$ seen by autoconversion, radiation, and settling on LWP is nearly identical to the slope inferred by subtracting the slope in the scavenging-only run from the control run following Eq. 11 (Fig. 7f).

Autoconversion and settling are likely to interact with each other in a non-linear manner. As one is suppressed as a sink of cloud liquid, the other will grow (for example, if the conversion of cloud to rain is rendered inefficient then the gravitational settling of cloud droplets will become more efficient as the liquid content grows). Analysis of cross talk between these terms is not the focus of this article, but sensitivity of LWP to changes in each mechanism is shown in Fig. 7f. The effect of perturbing settling $N_d$ on LWP is larger than the effect of perturbing the autoconversion $N_d$, singling it out as an important lever in controlling adjustment strength for future study.

Given that the scavenging-only simulation can be used to infer the adjustment strength in the UM, we also attempt to use the scavenging-only simulation to infer the strength of adjustments in the real world (Fig. 7d). This assumes that the statistical relationship between LWP and rain rates in the UM GA7.1, and the scavenging of CCN-relevant aerosol by precipitation in the UM GA7.1 are realistic.

    Based on the covariance between $N_d$ and LWP in the scavenging-only simulations in the UM GA7.1 and observations we offer an estimate of adjustment strength. The adjustment strength inferred from observations is stronger than

the UM for low SST and high WVP. Unlike the UM, weak negative covariance between LWP and $N_d$ exists for high SST and low WVP. This may indicate that the scavenging of aerosol by the UM is too efficient, or some other confounder of the relationship between LWP and $N_d$ is poorly represented in the model, or it may point to thinning of cloud via enhanced entrainment (Ackerman et al., 2004), which is not represented in the UM. Overall, the effect of increasing $N_d$ in the UM is to increase LWP.

We have discussed how to infer adjustment strength from the total covariance between $N_d$ and LWP, which mixes covariance induced by causality flowing from cloud to aerosol and aerosol to cloud. In the next section we show how this inferred adjustment strength from the simple regression model is able to predict LWP adjustments in response to anthropogenic aerosols within the full GCM simulation.

**3.4 Reproducing simulated adjustments between the pre-industrial and present day**

    We have produced an estimate of the sensitivity of LWP to changes in $N_d$ in the UM GA7.1 model (Fig. 7e). Does this simple regression model hold skill in reproducing the actual change in LWP within the model between the present day (PD) and pre-industrial (PI) aerosol emissions? Simulations are carried out setting aerosol emission to pre-industrial values. This is done for simulations using the GA7.1 scavenging parameterization. A second set of simulations is conducted where

the ECHAM-HAM5 scavenging is used to estimate the sensitivity to the representation of scavenging. PI, PD, and scavenging-only simulations are conducted for each scavenging configuration.

    The change in LWP between the PI and PD ($\Delta LWP_{PD-PI}$) simulated by the UM is 3.6 g/m$^2$ averaged across the 30-60°N region. Using the UM-simulated change in $N_d$ ($\Delta N_{dPD-PI}$) and the slope relating ln($N_d$) to LWP diagnosed from present-



day variability (Fig. 7) we calculate $\Delta LWP_{PD-PI}$. If the covariance between $\ln(N_d)$ and LWP observed in the control run (Fig. 7b) of the model and $\Delta N_{dPD-PI}$ are used to predict $\Delta LWP_{PD-PI}$, the negative correlation introduced by precipitation scavenging leads to a negative $\Delta LWP_{PD-PI}$ (Fig. 8a, in disagreement with the UM-simulated $\Delta LWP_{PD-PI}$). If the covariance in the scavenging-only simulation (causality flows from clouds to precipitation to aerosol to $N_d$) is used (Fig. 7c), the regression model-predicted decrease in LWP between PI and PD aerosol levels doubles. However, if we difference the sensitivity in the control and scavenging-only simulation to yield the sensitivity of LWP to changes in $N_d$ (Fig. 7e), the $\Delta LWP_{PD-PI}$ predicted by the regression model and the UM-simulated $\Delta N_{dPD-PI}$ agrees well with the UM-simulated $\Delta LWP_{PD-PI}$ (Fig. 8). This result supports the utility of the inferred sensitivity of LWP to $N_d$ in examining adjustments due to PD aerosol.

What does the covariance between LWP and $\ln(N_d)$ in the real world tell us about the model? Use of the observational estimate of adjustment strength (Fig. 7d) calculates a $\Delta LWP_{PD-PI}$ that is two thirds of the $\Delta LWP_{PD-PI}$ inferred from the control run and $\Delta N_{dPD-PI}$ (Fig. 8). This is because, while the inferred sensitivity of LWP to $N_d$ from observations is quite strong in some regions, it is weakly negative in the most commonly-occurring WVP-SST regimes (Fig. 7d). That is to say, if the efficacy of precipitation in removing CCN-relevant aerosol in the real-atmosphere is near to the efficacy in UM, the relationship between LWP and rain rates in the real atmosphere is near to the relationship in the UM, and the covariability between $N_d$ and LWP induced by other confounders of the $N_d$-LWP relationship are realistic, then the adjustments ($\Delta LWP_{PD-PI}$) simulated by the UM are not unreasonable, albeit a little large. This result is consistent with constraints provided by simulation of the Holuhraun eruption (Malavelle et al., 2017), which showed that the anomaly predicted by HadGEM3 was within the observed range. However, the observationally-inferred sensitivity of LWP to changes in $N_d$ is stronger in low SST, and high WVP regimes than it is in the UM GA7.1, but much weaker, or even slightly negative in low WVP and high SST regimes. This results in a strong latitudinal gradient in $\Delta LWP_{PD-PI}$ between 30°N and 60°N inferred from observations (Fig. 8b).

The predicted $\Delta LWP_{PD-PI}$ based on the control and scavenging-only simulations (Fig. 7e) underestimates the UM-simulated $\Delta LWP_{PD-PI}$ by around 25-35%. This may result from disregarding feedbacks between precipitation suppression and $N_d$ (Jing and Suzuki, 2018), or may simply be due to shortcomings in the simple linear model relating $\ln(N_d)$ to LWP. However, the fractional reduction in inferred $\Delta LWP_{PD-PI}$ when observations are used to constrain the sensitivity of LWP to $N_d$ suggests that $\Delta LWP_{PD-PI}$ should be around 70% of the value predicted by the UM GA7.1 averaged across the NH. $\Delta LWP_{PD-PI}$ inferred from observations is extremely dependent on latitude with almost no change in LWP over lower latitudes and warmer SSTs. We hasten to note that the change in $N_d$ between the PI and PD may not be perfectly reflect the real change in $N_d$, making the absolute values of the constrained $\Delta LWP_{PD-PI}$ less relevant than the fractional overestimation in UM-simulated $\Delta LWP_{PD-PI}$.

We have focused on changes in LWP in this work because it is a variable that we have good observations of, it can be compared between models and observations in a straight-forward way, and it clearly links to adjustments. However, this variable is not the key variable in discussing radiative forcing and climate sensitivity. How does the radiative forcing from adjustments scale with LWP? Determining the precise effects of adjustments on shortwave cloud radiative effect is difficult,





but if we assume that the perturbations in LWP induced by adjustments are similar to the perturbations that are driven by meteorology we can offer a simple estimate to inform our understanding of the modeled forcing from adjustments.

Examination of the relationship between LWP and albedo within the UM GA7.1 shows a rapid climb in albedo for low LWP, followed by saturation at higher values, as expected from saturation of cloud fraction and optical depth (Fig. 9). This curve can be fit by a second-degree polynomial, meaning that the sensitivity of albedo to changes in LWP is a function of the LWP. That is to say, regions where LWP is already high are going to have a smaller increase in albedo for a unit change in LWP. The change in LWP between PI and PD is much smaller than the range of the variation of LWP in the climate mean-state, so we use the average of monthly-mean LWP between PI and PD to calculate the sensitivity of albedo to changes in LWP for the NH midlatitudes. This sensitivity ($d\alpha/dLWP$) is multiplied by the change in LWP to yield change in albedo. This change in albedo is scaled by downwelling shortwave to give the change in reflected shortwave.

The change in reflected shortwave predicted from LWP changes in the UM GA7.1 is 1.9-2.0 Wm$^{-2}$, depending on the scavenging parametrization used. The change in shortwave inferred from the regression model of LWP trained in the control run and corrected by the scavenging-only run is 1.5 Wm$^{-2}$. If the observed sensitivity of LWP to $N_d$ is used to constrain $\Delta$LWP, the predicted change in reflected shortwave is approximately 1.0 Wm$^{-2}$. Thus, we estimate that GA7.1 overpredicts the change in reflected shortwave due to adjustments in response to a given change in $N_d$ by around 50%. This estimate is subject to the caveat that changes in LWP due to adjustments may not affect albedo in the same way as suggested by examining the total variability. For example, adjustments might only increase the liquid content of the very thickest clouds and have a relatively slight impact on albedo. However, it is unclear how to provide a more complex calculation than presented here.

## 4 Discussion

As the possible range for the radiative forcing from the first indirect effect has narrowed aerosol-cloud adjustments have become an increasingly central source of uncertainty in aerosol-cloud radiative forcing (Bellouin et al., 2019). Here we focus on the northern midlatitudes, where the majority of anthropogenic aerosol is emitted (Myhre et al., 2013). Previous work has examined aerosol-cloud adjustments in midlatitude cyclones, showing that cyclone liquid water path (LWP) increases with cloud droplet number concentration ($N_d$) (McCoy et al., 2018c). This work focuses on the remainder of cases in the midlatitudes when there is no cyclone center within 2000 km (roughly 42% of the time between 30°N-60°N, see Fig. 1).

Untangling the effect of cloud microphysics on cloud macrophysics from the total variability in cloud macrophysics is challenging. In the observational record we can only characterize the covariance between predictors and predictands. In interrogating the observed covariances between cloud properties and different meteorological predictors we find many of the relationships that have been described in the literature documenting cloud controlling factors (Myers and Norris, 2015).





Once meteorological variability is accounted for, statistically significant relationships between cloud microphysics ($N_d$) and cloud macrophysics (LWP) appear. In keeping with previous studies (Gryspeerdt et al., 2019), LWP and $N_d$ are found to be negatively correlated. However, this is clearly not consistent with the UM GA7.1 GCM's actual response in LWP to changes in aerosol emissions between PI and PD. This negative correlation is due to a combination of $N_d$ being driven by scavenging, as shown in Wood et al. (2012), and spurious correlation between $N_d$ and LWP driven by external variables affecting both terms (Mauger and Norris, 2007) overwhelming any positive covariance driven by aerosol cloud adjustments.

While we cannot disentangle the covariance due to adjustments and covariance due to scavenging and other confounders in the observational record, we can create simulations in which causality is forced to flow from clouds to Nd (scavenging-only). Using this measure of non-adjustment-induced variability in $N_d$, we can infer the effect of changing Nd on LWP. This inferred sensitivity of LWP to $N_d$ agrees well with simulations where $N_d$ is manually varied in the cloud microphysics (Fig. 7f). The inferred sensitivity of LWP to $N_d$, combined with the UM-predicted change in Nd reproduces the UM-predicted change in LWP between the PI and PD ($\Delta LWP_{PD-PI}$, Fig. 8). These two tests support this method's relevance to understanding aerosol-cloud adjustments.

The analysis presented here, combined with the enhancement in LWP in cyclone systems shown in McCoy et al. (2018c), points toward an overall increase in LWP across the NH midlatitudes in response to anthropogenic aerosol. Ultimately, while the regimes examined here, and in McCoy et al. (2018c) are very different, the detection of a change in LWP in response to changes in $N_d$ rests on the ability of the technique to account for non-adjustment-induced variability in $N_d$. In McCoy et al. (2018c) this was done by stratifying the dataset by cyclone precipitation rate, which is well-constrained by the large-scale environment (Field and Wood, 2007), which in turn stratifies the data set by the scavenging of aerosol.

Assuming that scavenging in the UM GA7.1 model is realistic, that the relationship between LWP and precipitation is reasonable, and that the non-causal covariance produced by other factors is replicated in the model, we evaluate the strength of adjustments based on observations. This reveals that the present version of the UM GA7.1 overestimates the sensitivity of LWP to changes in $N_d$ by approximately 50% outside of midlatitude cyclones. Calculation of the implied change in shortwave shows a similar overestimate in forcing due to adjustments by the UM. Observations also imply that aerosol-cloud adjustments in the UM GA7.1 are occurring in the wrong regime. Adjustments in the UM favor warmer SSTs and lower WVP, while the observations favor colder SSTs and higher WVP. This difference in regime may be due to early precipitation onset in the UM GA7.1.

Our result is in contradiction with previous empirical constraint studies that have postulated that changes in $N_d$ greatly enhance LWP (Rosenfeld et al., 2019), have little effect (Malavelle et al., 2017; Toll et al., 2017), or reduce LWP (Sato et al., 2018; Gryspeerdt et al., 2019). We suggest that this diversity within the literature is because a range of constraints may be arrived at depending on the degree to which precipitation scavenging and meteorological driving of aerosol and cloud occludes aerosol-cloud adjustments and what steps are taken in the analysis to account for scavenging-induced and meteorologically-induced covariability. Based on the analysis presented here we believe that positive, zero, or



extremely strongly negative radiative forcings due to aerosol-cloud adjustments in the midlatitudes are not supported by the observations.

**Acknowledgements**

DTM and PRF acknowledge support from the PRIMAVERA project, funded by the European Union's Horizon 2020
5    programme, Grant Agreement no. 641727. GE acknowledges support from the NASA MEaSUREs program (via subcontract with the Jet Propulsion Laboratory; Grant no. GG008658). We acknowledge use of the MONSooN system, a collaborative facility supplied under the Joint Weather and Climate Research Programme, a strategic partnership between the Met Office and the Natural Environment Research Council.

*Data availability*

MERRA-2 data were downloaded from the Giovanni data server
(https://disc.gsfc.nasa.gov/datasets/M2T1NXSLV_V5.12.4/summary?keywords=merra-2; last access: 22 July 2019). CERES data were downloaded through the ordering interface at https://ceres.larc.nasa.gov/order_data.php (last access: 22 July 2019).

*Author contributions*
All authors contributed ideas and helped edit the paper. DTM and PRF planned the paper. Data analysis and writing were undertaken by DTM. Present day and pre-industrial simulation suites were set up by HG. Simulations were run by DTM. Idealized simulations were set up by DTM. Development of simulations was conducted by HG, PRF, DPG, and DTM.
20   MODIS data was created by DPG. MAC-LWP data was created by GSE.

*Competing interests*
The authors declare that they have no conflict of interest.





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



**Table 1 List of observed variables.**

| Variable | Description | Source |
|---|---|---|
| LWP | Microwave liquid water path averaged over cloudy and clear sky. | (Elsaesser et al., 2017) |
| $N_d$ | MODIS cloud droplet number concentration. | (McCoy et al., 2018c; Grosvenor and Wood, 2014) |
| SST | OSTIA sea surface temperature as used by MERRA2 reanalysis as a boundary condition. | (Molod et al., 2015) |
| WVP | Column water vapor path as observed by MAC-LWP. | (Elsaesser et al., 2017) |
| $\omega_{550}$ | 550 hPa subsidence calculated by MERRA2. | (Molod et al., 2015) |
| EIS | Estimated inversion strength calculated following Wood and Bretherton (2006) using MERRA2 data. | (Molod et al., 2015; Wood and Bretherton, 2006) |
| SHF | Sensible heat flux calculated from MERRA2 data following Miyamoto et al. (2018). | (Molod et al., 2015) |



**Table 2 List of simulations. The MetOffice model designation, a short description, the scavenging coefficients used, the value of $N_d$ used in the radiation, settling, and autoconversion parameterizations, and the nudging (winds, or temperature). A tilde in the $N_d$ for a given parameterization indicates that $N_d$ was not set to a constant value.**

| Model label | Description | Scavenging | Radiation $N_d$ [cm$^{-3}$] | Settling $N_d$ [cm$^{-3}$] | Autoconversion $N_d$ [cm$^{-3}$] | Nudging |
|---|---|---|---|---|---|---|
| u-bh800 | Control | UM | ~ | ~ | ~ | U |
| u-bi580 | Control | ECHAM | ~ | ~ | ~ | U |
| u-bi674 | Scavenging-only | UM | 75 | 75 | 75 | U |
| u-bi677 | Scavenging-only | ECHAM | 75 | 75 | 75 | U |
| u-bh721 | PI aerosol | UM | ~ | ~ | ~ | U,T |
| u-bh722 | PD aerosol | UM | ~ | ~ | ~ | U,T |
| u-bi780 | PI aerosol | ECHAM | ~ | ~ | ~ | U,T |
| u-bi781 | PD aerosol | ECHAM | ~ | ~ | ~ | U,T |
| u-bi239 | Cloud microphysics $N_d$ =30 | UM | 30 | 30 | 30 | U |
| u-bi971 | Cloud microphysics $N_d$ =100 | UM | 100 | 100 | 100 | U |
| u-bi972 | Cloud microphysics $N_d$ =300 | UM | 300 | 300 | 300 | U |
| u-bi284 | Autoconversion $N_d$ =100 | UM | 30 | 30 | 100 | U |
| u-bi285 | Autoconversion $N_d$ =300 | UM | 30 | 30 | 300 | U |
| u-bi248 | Settling $N_d$ =100 | UM | 30 | 100 | 30 | U |
| u-bi250 | Settling $N_d$ =300 | UM | 30 | 300 | 30 | U |
| u-bi283 | Radiative transfer $N_d$ =100 | UM | 100 | 30 | 30 | U |
| u-bi282 | Radiative transfer $N_d$ =300 | UM | 300 | 30 | 30 | U |





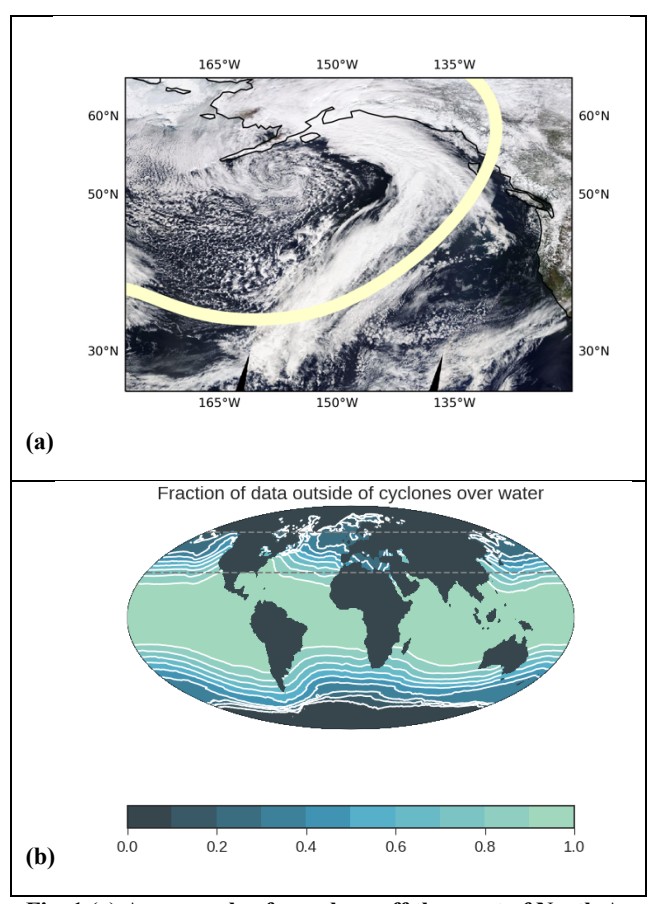

**Fig. 1 (a) An example of a cyclone off the coast of North America from MODIS Aqua. The cyclone center has been identified south of the Aleut Peninsula based on MERRA2 sea-level pressure (SLP). A line shows the edge of the area considered to lie within the cyclone following Field and Wood (2007). (b) The fraction of data outside of cyclones averaged between 2003 and 2015. The mean within the 30°N-60°N region is 42%.**

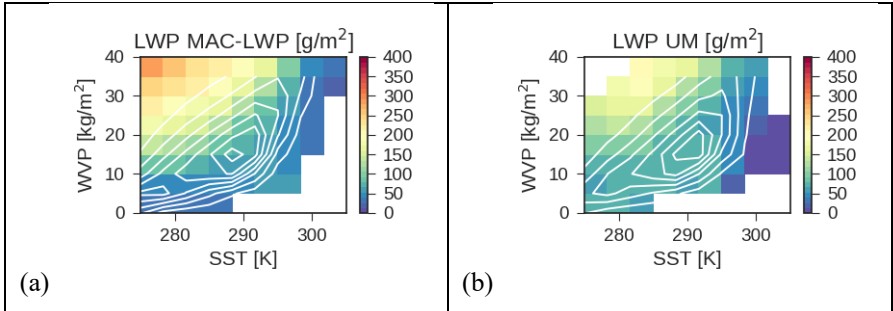

**Fig. 2 MAC-LWP observed (a) and UM-simulated (b) LWP in the space of SST and WVP in the region 30°-60°N. White lines show the distribution of data in SST-WVP space.**





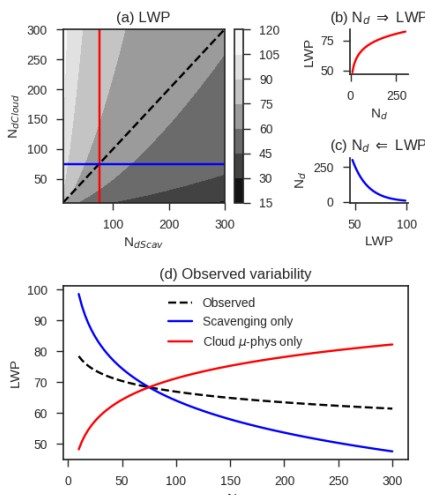

**Fig. 3 A simple conceptual model demonstrating how variance due to precipitation-driving of $N_d$ and due to $N_d$ driving changes in LWP through cloud microphysics affects total covariance between $N_d$ and LWP in observations and models. $N_d$ where scavenging is the only causal link between cloud and aerosol ($N_{dScav}$) and $N_d$ affecting the cloud microphysics ($N_{dCloud}$) are considered as distinct entities. The contour map in (a) illustrates a potential topological LWP space in terms of $N_{dCloud}$ and $N_{dScav}$. The case where $N_{dCloud}$ is set to a constant value is shown with a blue line and where $N_{dCloud}$ is increased independently is shown with a red line. In a causal sense this implies $N_{dCloud}$ driving LWP (shown in b) and $N_{dScav}$ being driven by LWP (shown in c), where $N_{dScav}$ is negatively correlated with LWP because $dN_{dScav}/dR < 0$ and $dLWP/dR > 0$, where R is rain rate. The observed variability of LWP in terms of $N_d$ is shown in (d) for the cases $N_{dCloud} = N_d$ and $N_{dScav} = constant$ (red); $N_{dScav} = N_d$ and $N_{dCloud} = constant$ (blue); and $N_{dCloud} = N_{dScav} = N_d$ (the real world, dashed line).**



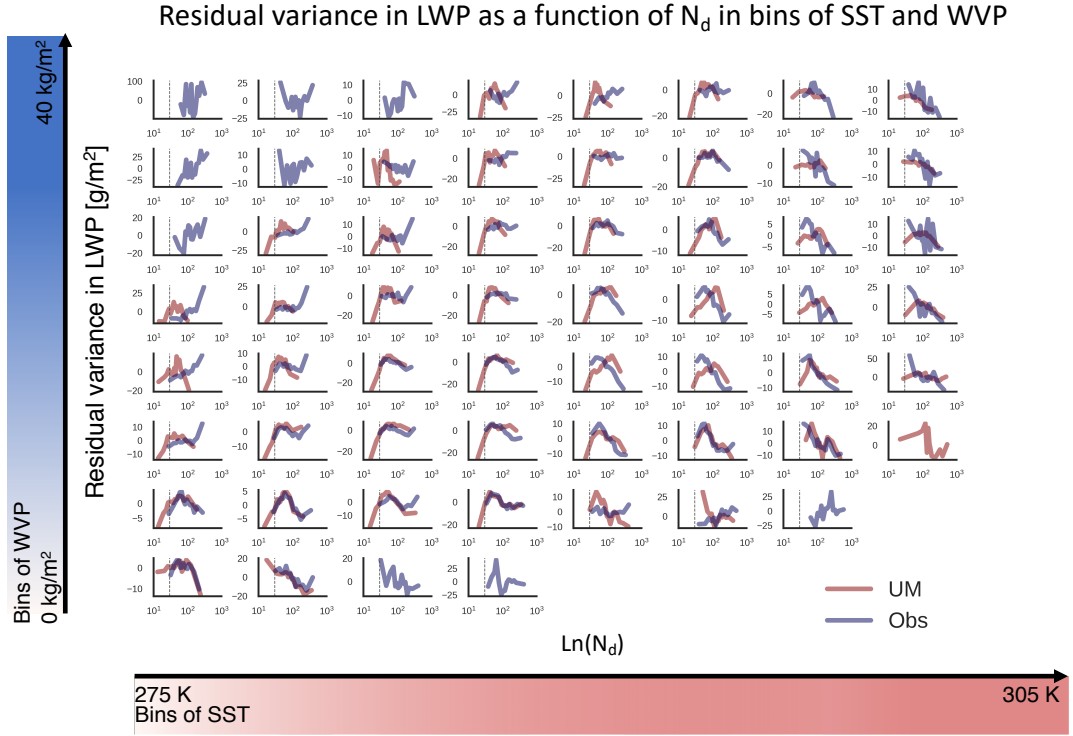

**Fig. 4 Residual variance in LWP after removing variability associated with other predictors (Eq. 12) plotted as a function of $N_d$ in the UM GA7.1 and observations. Each subplot corresponds to a bin of WVP and SST. Due to retrieval limitations, values of $N_d$ below 30 cm$^{-3}$ (dashed grey line) are almost never observed, but occur in model output.**





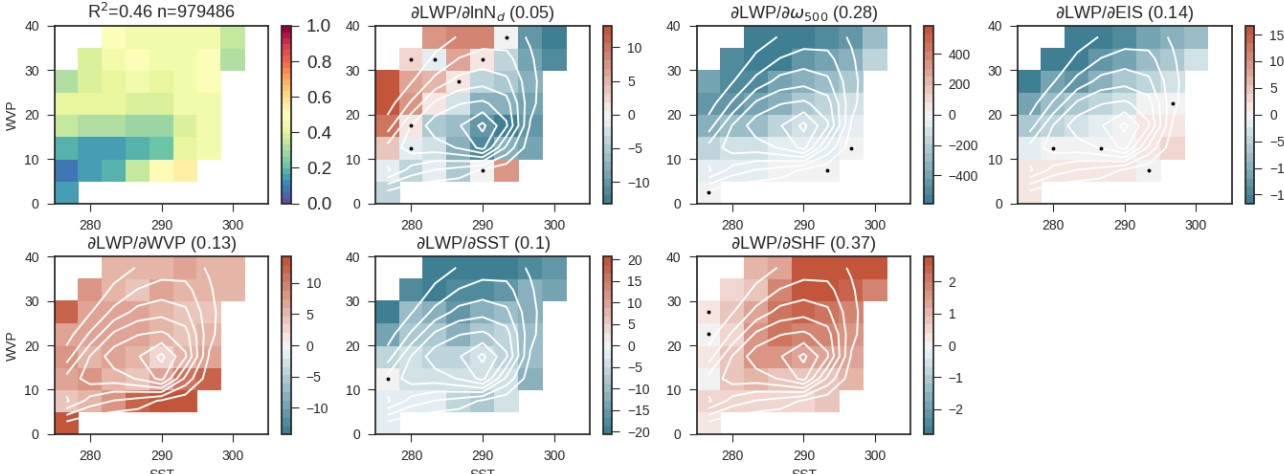

**Fig. 5 The regression coefficients relating observed LWP to predictors in the phase space of SST and WVP. Regression coefficients not significant at 99% confidence are marked with a dot. White contours show the distribution of observations in the WVP-SST phase space. The first plot shows the explained variance in each bin. The explained variance within the entire data set is noted in the title along with the number of observations. The remaining plots show the coefficients relating each predictor to LWP. In the title the weighted mean of the absolute value of the standardized coefficient ($|\partial \sigma LWP / \partial \sigma x|$) is shown in brackets to give an estimate of contribution of each predictor to the variance. Regression coefficients are as described in Eq. 4.**





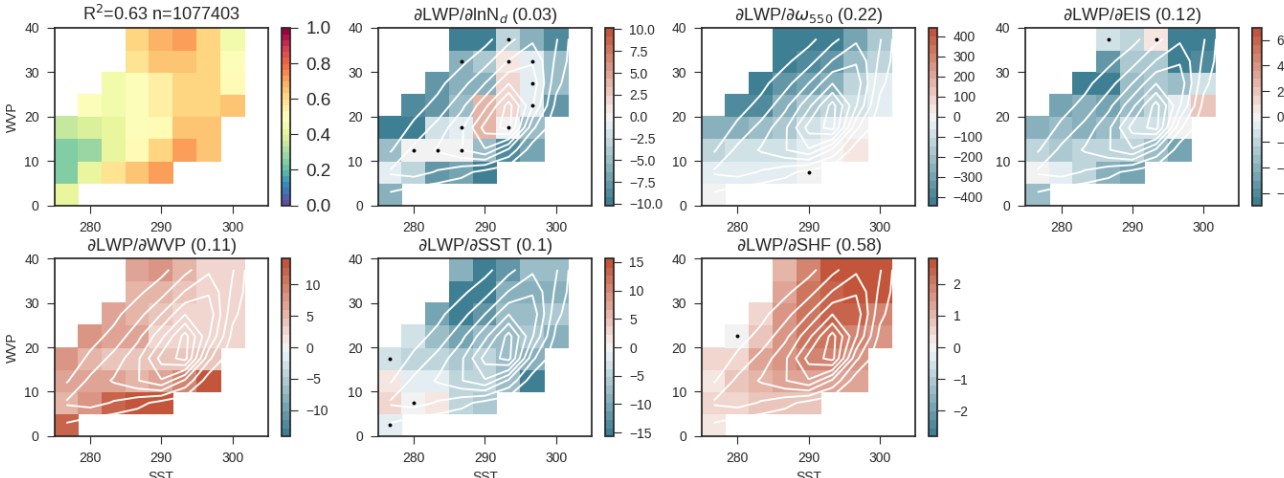

Fig. 6 as in Fig. 5, but showing the covariance within the UM.

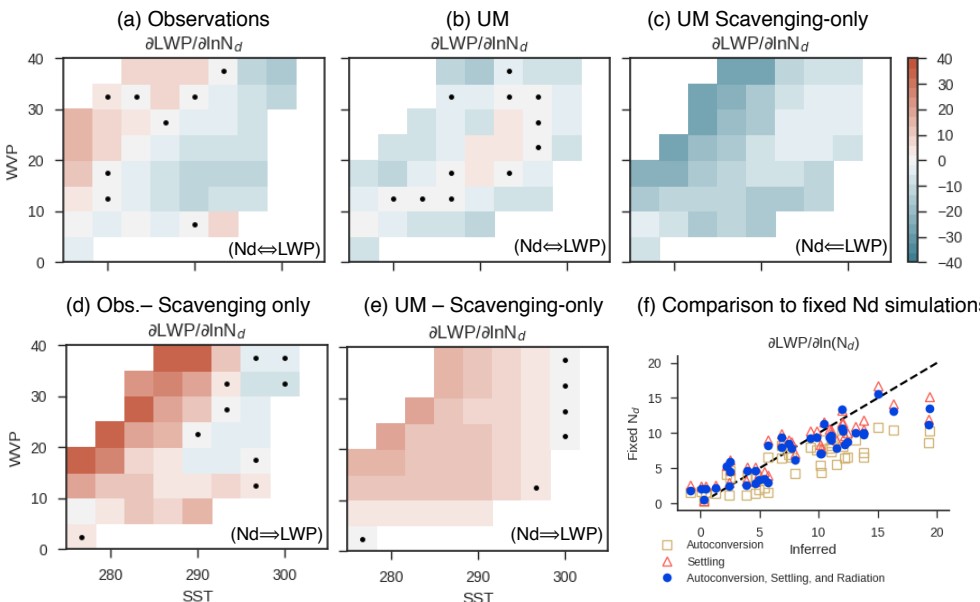

**Fig. 7 The observed covariance between $N_d$ and LWP and the inferred effect of $N_d$ on LWP. Notations in the bottom right of plots indicate the expected direction of causality. The covariance between $N_d$ and LWP is shown in (a) the observational record (as in Fig. 5); (b) the control simulation of the UM GA7.1 (as in Fig. 6), and (c) a simulation where the $N_d$ seen by the autoconversion, settling, and radiation is set to 75 cm⁻³ and only scavenging links $N_d$ and LWP. The inferred strength of adjustments in observations (d) calculated as the difference in slope between (a) and (c) as in Eq. 11, and in (e) the UM calculated as the difference in slope between (b) and (c). Inferred and true adjustment strength for the UM are compared in (f). The inferred adjustment strength in (e) is compared to the sensitivity of LWP to $N_d$ in a set of simulations where $N_d$ is fixed in the autoconversion, settling, and radiation to a range of values, forcing causality to flow from $N_d$ to LWP. The one-to-one line is shown with dashes. Different symbols indicate whether the $N_d$ seen by the autoconversion, settling, or radiation was varied while others were held constant. Each symbol corresponds to a bin in WVP and SST, as in (e).**



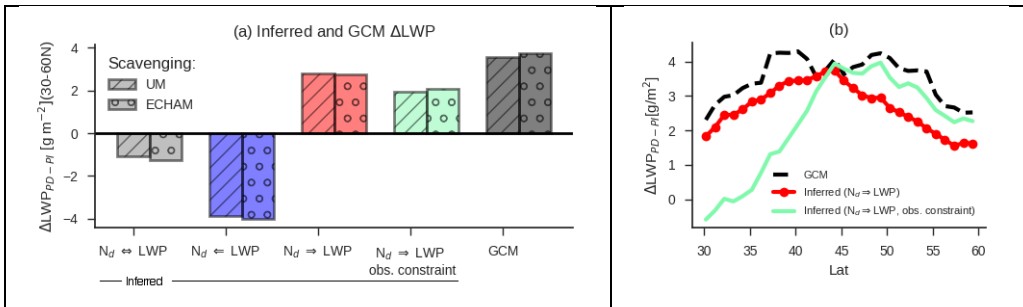

5   **Fig. 8 The change in LWP between the PI and PD aerosol emissions simulations ($\Delta$LWP$_{PD-PI}$) inferred by covariance between $N_d$ and LWP in the PD, and as simulated by the UM. The inferred $\Delta$LWP$_{PD-PI}$ is calculated using the UM-simulated change in $N_d$. Subplot (a) shows the inferred $\Delta$LWP$_{PD-PI}$ based on variance in different simulations. The direction of causality is indicated for each case. From left to right: $\Delta$LWP$_{PD-PI}$ inferred from covariance in the control simulation ($N_d \Leftrightarrow LWP$, causality between $N_d$ and LWP goes both directions Fig. 7b), inferred from the scavenging-only simulation ($N_d \Leftarrow LWP$, causality from LWP to $N_d$ Fig.**

10  **7c), inferred by correcting the total covariance using covariance in the scavenging-only simulation ($N_d \Rightarrow LWP$, causality goes from $N_d$ to LWP Fig. 7d - this is the adjustment strength in UM-GA7.1 inferred by the method presented in this paper), and when the covariance in observations is combined with the scavenging-only simulation (listed as obs. constraint). The $\Delta$LWP$_{PD-PI}$ simulated by the GCM is shown on the right in black. This is the true aerosol-cloud adjustment in the GCM that is compared to the value being inferred in the UM GA7.1 as shown by the red bars. $\Delta$LWP$_{PD-PI}$ is provided for simulations using the UM GA7.1**

15  **scavenging and ECHAM-HAM5 scavenging. In (b), as in (a), but resolved in latitude and only showing the case when the UM GA7.1 default scavenging is used.**

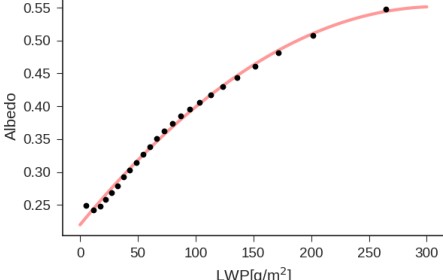

**Fig. 9 UM GA7.1 daily-mean albedo as a function of LWP over ocean between 30-60°N and outside of cyclones in equal quantiles of LWP (black circles). A second-order polynomial fit is shown using a red line.**

