# Peer review of "Untangling causality in midlatitude aerosol-cloud adjustments"

_Atmospheric Chemistry and Physics, 2019_

## Referee Comment (RC1) · Anonymous Referee #1 · 18 Sep 2019

**Review for: *"Untangling causality in midlatitude aerosol-cloud adjustments"* by McCoy et al submitted to ACPD**

Summary:

The authors address the issue of covariability between LWP and Nd that is not representative of cloud adjustments and provide an estimate of the radiative forcing due to cloud adjustments in Northern Hemisphere low clouds based on UM simulations constrained by observations. In particular, the authors focus on the fact that Nd variability is not only generated through variability in aerosol and cloud microphysical processes, which then may impact LWP, but that vice versa large LWP are associated with an increased likelihood in precipitation and thus scavenging of Nd. The latter may also contribute substantially to overall Nd variability in low clouds of the Northern Hemisphere, but is not representative of cloud adjustments to a perturbation in aerosol concentrations. To address this question the authors use a combined approach of remote sensing cloud retrievals and climate model simulations. The authors further argue that their simulation approach generally allows one to remove the Nd-LWP covariance representative of cloud adjustments from other compounding factors in addition to precipitation scavenging. They show that following their approach the sign and magnitude of the estimated cloud adjustment is impacted in global climate model simulations of present-day and preindustrial aerosol conditions.

General Comments:

This paper suggests a new approach to quantify the cloud adjustment in low-level clouds over the ocean and in particular, the Northern Hemisphere. The authors address one of the key scientific challenges in the quantification of the effective radiative forcing of anthropogenic aerosols and thus this paper is an excellent fit for this journal. The paper is well written, comprehensive, and I appreciate the careful discussion of correlation and the inference of causality throughout the manuscript. Prior to publication, I would like further clarification on their Ansatz as outlined below.

Specific comments:

- I am not sure I follow the author's argumentation for $Nd = Nd\_cloud = Nd\_scav$. Based on your simulation strategy, you can have $Nd = Nd\_scav$, or $Nd = Nd\_cloud$ by decoupling individual processes from Nd variability. However, the observed variability in Nd will always be a combination of the two. That is, a fraction of the change in Nd is due to changes in Nd that could impact LWP and the remaining fraction is representative of clouds impacting Nd + environmental factors.
  I agree with your point that some of the signal (or perhaps a large fraction) in the LWP-Nd relationship inferred from observations may be misinterpreted as a negative LWP adjustment, which is in truth driven by scavenging processes. However, I don't follow the logic from Equ. 8 to Equ. 9. In my mind, the terms $dlnNd\_scav/dlnNd$ and $dlnNd\_cloud/dlnNd$ do not always equal 1, but range between 0 and 1. Please clarify.

- You nudge winds and temperature down to 1.7km altitude, yet analyse boundary layer clouds up to 3.2km in altitude. Can you be sure that your nudging has no impact on your derived relationships? And why do you nudge different variables for the different model experiments (Table 2)?

- Equ. 4: Have you performed tests on overfitting? For instance what is the reasoning for fitting both EIS and $omega\_550$? Is there really additional skill added to the fit by including both?

- P8L7ff: I found this sentence confusing (see general comment above). Isn't it rather that precipitation scavenging is the only source of Nd variability? And that this variability is not representative of cloud adjustments?

- P9L25 & Fig 4. "there is significant variability in LWP associated with Nd". This is only true if the full residual LWP-LWP_fitted is entirely flat and has no remaining functional dependence (i.e. your predictors describe all of the variability, which may not be the case). Have you checked for this?

- P12L7: It is interesting that you find the settling Nd to have the dominant effect over autoconversion. I would argue that this deserves further comment. Is there any support from this from observations? If it would be specific to the UM model, how would that affect your conclusions?

Typos & Text edits:

- P2L17: suggest rephrase "Because of this constraining the..." to "Because of this, constraining the..."
- P2L27: suggest rephrase "Simulation of deep..." to "A simulation of deep..."
- P3L18: rephrase "have an overall" to "has an overall"
- P4L13: rephrase "is explain" to "is to explain"
- P13L28: rephrase "may not be perfectly reflect"
- Table 2: Simulation acronyms not clear
- Fig. 2: What is the contour spacing of the white contours?

---

## Referee Comment (RC2) · Anonymous Referee #2 · 24 Sep 2019

In this study, McCoy et al. used satellite observations and a GCM to quantify the LWP response to Nd in areas between NH midlatitude cyclones. A multi-variate linear regression model is used to reveal the covariance between LWP and other cloud-controlling factors. By performing sensitivity tests using a GCM, they tried to isolate the LWP response to aerosols due to scavenging from other microphysical processes. This approach provides a process-oriented analysis framework to evaluate the model. Overall I find the study very interesting, the results are well-supported by the analysis, and the presentation is good. So, I recommend publication in ACP.

I have several comments/questions for the authors to consider to improve the clarity of the paper:

1. Page 2, line 19-21: The sentence does not read well. The first half of the sentence

stated that aerosol-cloud interactions play a minor role, but the second half of the sentence stated the importance of making data volume available. I don't understand the logic here. Moreover, it might be necessary to substantiate the statement "Aerosol-cloud interactions are likely to contribute a relatively small fraction of overall variability in cloud properties" by providing some references.

2. This study focuses on the interstitial regions between cyclones only, but there are many other cloud regimes where aerosols can have similar or different impacts on clouds. It might be helpful to add a paragraph to discuss the applicability as well as limitations of this study.

3. Page 4, line 13: add "to" before "explain"?

4. What is the effect of the degradation of spatial resolution (when all the data is regridded to 1-degree) on the covariance analysis? Is ACI derived from the gridded data still representing the physics? Could you discuss if this is an important issue (as discussed in McComiskey and Feingold, 2012) that affects the conclusion of the study?

5. Page 6, line 13-14: Do you mean aerosol emissions or concentrations are set to PI and PD for the two simulations? If emissions, please provide the reference for the emission data. If concentrations, please provide the reference for the concentration data.

6. Section 2.5, I understand that you intend to separate the scavenging effect from other microphysical processes. However, these processes of course are nonlinear. Could you provide a discussion on the applicability and limitation of the linear assumption here as well as the use of multiple linear regression?

7. In Fig. 4, where LWP decreases with increasing Nd due to precipitation effects, do you mean in terms of grid-box mean LWP and Nd or in-cloud LWP and Nd? I think I understand the effect in terms of grid-box means but not sure I understand in terms of in-cloud values. Could you please clarify?

**[ACPD](ACPD)**

Interactive
comment

8. Page 14, line 22-23. This first sentence reads awkward. Please rephrase.

9. In discussion, the authors indicated that this study is in contradiction with several previous studies. Could you please discuss the disagreement? Is it because a different observational data, different GCM, different analysis framework, or different regime is used?

---

## Author Comment (AC1) · 27 Nov 2019

We thank the reviewers for their supportive and insightful comments. Responses and track changes are attached as a PDF.

Please also note the supplement to this comment:
https://www.atmos-chem-phys-discuss.net/acp-2019-665/acp-2019-665-AC1-supplement.pdf

---

## Author Response (AR1)

**Response to reviewers:**

We thank reviewer 1 and 2 for their careful reading of our paper and their supportive and helpful comments. This document details the changes made to the manuscript. Reviewer comments are in blue. Responses are in red.

**Response to R1:**

I am not sure I follow the author's argumentation for Nd=Nd_cloud=Nd_scav. Based on your simulation strategy, you can have Nd=Nd_scav, or Nd=Nd_cloud by decoupling individual processes from Nd variability. However, the observed variability in Nd will always be a combination of the two. That is, a fraction of the change in Nd is due to changes in Nd that could impact LWP and the remaining fraction is representative of clouds impacting Nd + environmental factors.

I agree with your point that some of the signal (or perhaps a large fraction) in the LWP-Nd relationship inferred from observations may be misinterpreted as a negative LWP adjustment, which is in truth driven by scavenging processes. However, I don't follow the logic from Equ. 8 to Equ. 9. In my mind, the terms dlnNd_scav/dlnNd and dlnNd_cloud/dlnNd do not always equal 1, but range between 0 and 1. Please clarify.

You bring up a very good point and this section was poorly explained. It might be better to portray 'Nd' in this case as the Nd in the regression of LWP on Nd that is used to characterize covariance between LWP and Nd. In the case where Nd=Nd_cloud we run many simulations where we replace the Nd in the autoconversion, settling, and radiation (ASR) with a constant. The regression analysis is performed using many simulations with constant Nd in the ASR, which is then used as the independent variable in the regression. In the case where Nd=Nd_scav we take one of these simulations where Nd in the ASR is set to a constant and regress on the Nd calculated by the nucleation scheme. In the last case where Nd=Nd_cloud=Nd_scav, this is just the control run where the Nd_cloud (eg the Nd in the ASR) is linked to the Nd calculated by the nucleation. In this case the Nd seen by the ASR, and driven by the scavenging are not distinct entities and Nd_cloud =Nd=Nd_scav so the derivative of one with respect to another goes to one (dy/dx =1 if y=x).

We have added additional text and a flow chart to the paper to try and clarify this calculation.

You nudge winds and temperature down to 1.7km altitude, yet analyse boundary layer clouds up to 3.2km in altitude. Can you be sure that your nudging has no impact on your derived relationships? And why do you nudge different variables for the different model experiments (Table 2)?

Also a great point. In the context of this calculation nudging the large-scale environment should only manifest through the inferred meteorological controls on cloud. Nudging was used in this study for two reasons: 1) because the simulations are relatively short so nudging creates more

comparable simulations for a shorter integration in the sense of looking at the same meteorological regimes 2) because we wanted to compare to the observations and wanted to try to sample similar meteorological regimes. You make a very good point about the nudging of U and T in the PI-PD simulations. This was copied from existing suites at the MetOffice used to calculate the aerosol radiative forcing(Yoshioka et al., 2019). Temperature nudging has now been turned off. This shows the same results qualitatively, but shifts the inferred and true ΔLWP between PI and PD closer together. Additional simulations were carried out with nudging starting at level 27 (~5km) instead of level 16 (~1.8km) of the model, and without wind nudging altogether. The results from these simulations are shown in the SM and do not qualitatively change the results of this study.

Equ. 4: Have you performed tests on overfitting? For instance what is the reasoning for fitting both EIS and omega_550? Is there really additional skill added to the fit by including both?

The use of both EIS and omega 550 was based on existing research showing that EIS and subsidence correlate, but have opposing effects in terms of their effect on low cloud cover (Myers and Norris, 2013; Wall et al., 2017). We have tested for overfitting by training our model in the control run and then predicting dLWP/dln(Nd) in simulations with very low and very high Nd (Fig. 8f of the main text).

P8L7ff: I found this sentence confusing (see general comment above). Isn't it rather that precipitation scavenging is the only source of Nd variability? And that this variability is not representative of cloud adjustments?

This is right except that Nd variability is also due to sources of aerosol, advection, and sinks like dry deposition. We have added a new schematic figure illustrating the links between cloud, aerosol, and meteorology and have added text to try and explain this better.

P9L25 & Fig 4. "there is significant variability in LWP associated with Nd". This is only true if the full residual LWP-LWP_fitted is entirely flat and has no remaining functional dependence (i.e. your predictors describe all of the variability, which may not be the case). Have you checked for this?

We have changed this sentence to reflect that we meant that the coefficient in the multiple linear regression is confidently greater than zero. There is no requirement for significance to mean that all variability is explained.

P12L7: It is interesting that you find the settling Nd to have the dominant effect over autoconversion. I would argue that this deserves further comment. Is there any support from this from observations? If it would be specific to the UM model, how would that affect your conclusions?

We also thought this was an interesting effect. Unfortunately it is hard to say how a metric similar to the precipitation probability Spop used in many studies to characterize the response of precipitation to aerosol(Ma et al., 2018) could be created to look at settling of cloud droplets. However, comparison to other model studies does show that in regions of low cloud with less frequent precipitation the process rate associated with settling is large compared to autoconversion (Gettelman et al., 2013), which opens the door to this process being a significant source of aerosol-cloud adjustments.

Overall, our results are insensitive to whether precipitation or settling suppression dominate adjustments as we neglect feedbacks between adjustments and scavenging. We have added text to clarify this. Thank you for suggesting this.

Typos and text edits have been changed as suggested. In regards to acronyms in Table 2 being unclear: the first column is just the unique identifier of the simulation in the MetOffice repository so that they can be requested by people wanting to replicate our analysis. Are these what you were referring to? We added some additional text to the descriptions to make it clearer what the simulation with PI and PD simulations are for. Thank you also for finding these.

R2:

Page 2, line 19-21: The sentence does not read well. The first half of the sentence stated that aerosol-cloud interactions play a minor role, but the second half of the sentence stated the importance of making data volume available. I don't understand the logic here. Moreover, it might be necessary to substantiate the statement "Aerosol- cloud interactions are likely to contribute a relatively small fraction of overall variability in cloud properties" by providing some references.

This was unclearly expressed. What we meant to convey is that detecting aerosol-cloud adjustments above the background of meteorological variability presents us with a signal-to-noise problem with the small amount of variability in LWP that is due to aerosol-cloud adjustments against the large amount of variability driven by dynamics and thermodynamics. To overcome small signal to noise ratio we need to use a lot of data to be able to detect a signal. We have changed the sentence to try and better communicate this. We have added a reference that puts an upper limit on the amount of variance explained by adjustments, and which is generally small.

2. This study focuses on the interstitial regions between cyclones only, but there are many other cloud regimes where aerosols can have similar or different impacts on clouds. It might be helpful to add a paragraph to discuss the applicability as well as limitations of this study.

Great point- we did a previous study in cyclones, which is noted in the introduction and in the conclusions. We have added a few sentences to expand on this and to discuss how adjustments might look in the tropics and subtropics and limitations in our analysis technique.

Page 4, line 13: add "to" before "explain"?

Changed.

What is the effect of the degradation of spatial resolution (when all the data is regridded to 1-degree) on the covariance analysis? Is ACI derived from the gridded data still representing the physics? Could you discuss if this is an important issue (as discussed in McComiskey and Feingold, 2012) that affects the conclusion of the study?

(McComiskey and Feingold, 2012) examined the first indirect effect (the Twomey effect), but it is still important to consider the effects of resolution on our results. It is reasonable to suppose that degradation of the spatial resolution will lead to weakening the multiple linear regression coefficients as a function of decreasing variance in the dataset. However, we validate our projected aerosol-cloud adjustment by comparing the change in LWP inferred from the variance in the PD of the GCM to the change in LWP between the PI and PD calculated by the GCM (Fig. 9b) as well as the response in simulations with prescribed Nd in the cloud microphysics (Fig. 8f). The change inferred by the 1x1° daily-mean UM GA7.1 data in the PD agrees with both these tests, neither of which are sensitive to resolution. We have added text discussing resolution.

5. Page 6, line 13-14: Do you mean aerosol emissions or concentrations are set to PI and PD for the two simulations? If emissions, please provide the reference for the emission data. If concentrations, please provide the reference for the concentration data.

It is aerosol and chemistry emissions. Anthropogenic emissions are the CMIP6 emissions(Eyring et al., 2016). Natural emissions are from MEGAN-MACC(Sindelarova et al., 2014) over land and POET(Granier et al., 2005) over ocean. This has been added to the document.

Section 2.5, I understand that you intend to separate the scavenging effect from other microphysical processes. However, these processes of course are nonlinear. Could you provide a discussion on the applicability and limitation of the linear assumption here as well as the use of multiple linear regression?

Additional discussion has been added to the end of Section 2.5 noting that the relationship between LWP and Nd appears to be approximately linear and discussing that in regimes or models with very large adjustments moderated through precipitation our analysis framework will break down.

Eyring, V., Bony, S., Meehl, G. A., Senior, C. A., Stevens, B., Stouffer, R. J., and Taylor, K. E.: Overview of the Coupled Model Intercomparison Project Phase 6 (CMIP6) experimental design and organization, *Geosci. Model Dev.*, 9, 1937-1958, 10.5194/gmd-9-1937-2016, 2016.

Gettelman, A., Morrison, H., Terai, C., and Wood, R.: Microphysical process rates and global aerosol–cloud interactions, *Atmospheric Chemistry and Physics*, 13, 9855-9867, 2013.

Granier, C., Lamarque, J. F., Mieville, A., Muller, J. F., Olivier, J., Orlando, J., Peters, J., Petron, G., Tyndall, G., and Wallens, S.: POET, a database of surface emissions of ozone precursors, 2005.

Ma, P.-L., Rasch, P. J., Chepfer, H., Winker, D. M., and Ghan, S. J.: Observational constraint on cloud susceptibility weakened by aerosol retrieval limitations, *Nature Communications*, 9, 2640, 10.1038/s41467-018-05028-4, 2018.

McComiskey, A., and Feingold, G.: The scale problem in quantifying aerosol indirect effects, *Atmos. Chem. Phys.*, 12, 1031-1049, 10.5194/acp-12-1031-2012, 2012.

Myers, T. A., and Norris, J. R.: Observational Evidence That Enhanced Subsidence Reduces Subtropical Marine Boundary Layer Cloudiness, *Journal of Climate*, 26, 7507-7524, 10.1175/JCLI-D-12-00736.1, 2013.

Sindelarova, K., Granier, C., Bouarar, I., Guenther, A., Tilmes, S., Stavrakou, T., Müller, J. F., Kuhn, U., Stefani, P., and Knorr, W.: Global data set of biogenic VOC emissions calculated by the MEGAN model over the last 30 years, *Atmos. Chem. Phys.*, 14, 9317-9341, 10.5194/acp-14-9317-2014, 2014.

Wall, C. J., Hartmann, D. L., and Ma, P.-L.: Instantaneous Linkages between Clouds and Large-Scale Meteorology over the Southern Ocean in Observations and a Climate Model, *Journal of Climate*, 30, 9455-9474, 10.1175/jcli-d-17-0156.1, 2017.

Yoshioka, M., Regayre, L. A., Pringle, K. J., Johnson, J. S., Mann, G. W., Partridge, D. G., Sexton, D. M. H., Lister, G. M. S., Schutgens, N., Stier, P., Kipling, Z., Bellouin, N., Browse, J., Booth, B. B. B., Johnson, C. E., Johnson, B., Mollard, J. D. P., Lee, L., and Carslaw, K. S.: Ensembles of Global Climate Model Variants Designed for the Quantification and Constraint of Uncertainty in Aerosols and their Radiative Forcing, *Journal of Advances in Modeling Earth Systems*, 0, 10.1029/2019ms001628, 2019.

[revised manuscript text omitted]

---

## Author Response (AR2)

Thank you for your improvements to the paper. Comments are in bold.
**Your references are not to standard format. Can you edit it so that**

**(a) journal names use the appropriate abbreviations**
**(b) the titles of articles all use sentence capitalization**
**(c) fix all spacing issues (some commas need spaces beforehand or afterwards)**
**(d) anything else you find**

Based on assistance from Anja Rasmussen at Copernicus I have corrected the journal abbreviations. Titles are now sentence capitalized. Anja confirms that all spacing issues in references are correct and that references all look ok. I have read through the paper and removed extraneous spaces in the acknowledgements and corrected typos throughout the main text.

**I'll ask you to consider adding a nomenclature/glossary section since there are a lot of different variables and abbreviations in the paper. I think it would make it easier to read.**

I am amenable to a nomenclature/glossary in the SM, but a glossary section for variables already exists in Table 1. What abbreviations were you seeing as needing a glossary that are not in table 1?

[revised manuscript text omitted]
 \left(N_{d\,Scav}\right)} d\ln \left(N_{d\,Scav}\right) + \frac{\partial LWP}{\partial \ln \left(N_{d\,Cloud}\right)} d\ln \left(N_{d\,Cloud}\right) \qquad [7]$$

We assume that scavenging is unaffected by changes in $N_{d_{Cloud}}$ affecting rain rates, so there are no higher-order terms, although this may not be the case in reality (Wood et al., 2012), and this feedback is found to substantially enhance the strength of adjustments in some models (Jing and Suzuki, 2018). A visualization of Eq. 7 is shown in Fig. 4a. The best fit

5   of LWP to $N_d$ gives the sum of the partial derivatives of the two terms

$$\frac{dLWP}{d\ln(N_d)} = \left(\frac{\partial LWP}{\partial \ln \left(N_{d\,Scav}\right)}\right)_{N_{dCloud}} \frac{d\ln \left(N_{d\,Scav}\right)}{d\ln(N_d)} + \left(\frac{\partial LWP}{\partial \ln \left(N_{d\,Cloud}\right)}\right)_{N_{dScav}} \frac{d\ln \left(N_{d\,Cloud}\right)}{
[revised manuscript text omitted]